# Tracking individual action potentials throughout mammalian axonal arbors

**Milos Radivojevic[†]\*, Felix Franke[†], Michael Altermatt, Jan Müller, Andreas Hierlemann, Douglas J Bakkum**

Department of Biosystems Science and Engineering, ETH Zurich, Basel, Switzerland

**Abstract** Axons are neuronal processes specialized for conduction of action potentials (APs). The timing and temporal precision of APs when they reach each of the synapses are fundamentally important for information processing in the brain. Due to small diameters of axons, direct recording of single AP transmission is challenging. Consequently, most knowledge about axonal conductance derives from modeling studies or indirect measurements. We demonstrate a method to noninvasively and directly record individual APs propagating along millimeter-length axonal arbors in cortical cultures with hundreds of microelectrodes at microsecond temporal resolution. We find that cortical axons conduct single APs with high temporal precision (~100 µs arrival time jitter per mm length) and reliability: in more than 8,000,000 recorded APs, we did not observe any conduction or branch-point failures. Upon high-frequency stimulation at 100 Hz, successive became slower, and their arrival time precision decreased by 20% and 12% for the 100th AP, respectively.
DOI: https://doi.org/10.7554/eLife.30198.001

## Introduction

According to the classical view, axons are seen as conduction cables that reliably transmit action potentials (APs) between synaptically connected neurons in a digital (all-or-none) fashion (*Hodgkin and Huxley, 1952*). However, more recently, a growing body of evidence suggest that a more refined view of axonal conduction might be appropriate. A number of experimental and modeling studies provided insights into different mechanisms that might fine-tune information processing in mammalian axons, and, thereby, affect synaptic transmission (*Debanne, 2004*; *Bucher and Goaillard, 2011*; *Debanne et al., 2011*; *Sasaki et al., 2011*). It has been shown that a modest increase in the somatic membrane potential of presynaptic cortical neurons leads to a proportional increase in the amplitude of axonal action potential (*Shu et al., 2006*). An increase in the AP amplitude facilitated synaptic release and increased the average amplitude of the postsynaptic potential (*Shu et al., 2006*; *Alle and Geiger, 2006*). In addition, it was found that gating dynamics of voltage-gated ion channels can regulate the length of axonal AP waveforms and, thereby, modulate the synaptic release. For example, slow inactivation of voltage-gated $K^+$ channels in mossy fiber boutons was found to mediate activity-dependent broadening of APs during high-frequency neuronal activity, which facilitated synaptic transmission (*Geiger and Jonas, 2000*). Contrary to a classical notion that information flows from the dendritic membrane toward the axon terminals, parallel patch-clamp recordings from soma and axons revealed that axons possess a certain level of functional independence from the presynaptic input. Thus, for example, during gamma oscillations of pyramidal cells, ectopic APs were found to be generated in the distal axon but did not invade the soma (*Dugladze et al., 2012*). Moreover, it has been shown that hippocampal interneurons can share the output across a coupled network of axons and can respond with persistent firing even in the absence of input to the soma or dendrites (*Sheffield et al., 2011*).

Moreover, the temporal precision with which APs arrive at a neurons' synapses is essential for encoding and computing neural information (*Rieke, 1999*). E.g., cortical neurons are specialized to

**\*For correspondence:**
rmilosh@gmail.com

[†]These authors contributed equally to this work

**Competing interests:** The authors declare that no competing interests exist.

detect coincident arrival of APs on the order of milliseconds (*Stuart and Häusser, 2001*). In contrast, theoretical studies suggest that ion channel noise (*Hodgkin and Huxley, 1952*; *Clay and DeFelice, 1983*; *White et al., 2000*) introduces random variability in axonal AP waveforms (*Neishabouri and Faisal, 2014*), decreases the temporal precision of AP propagation (*Faisal and Laughlin, 2007*), and compromises the reliability of the conduction itself (*Faisal and Laughlin, 2007*; *Skaugen and Walløe, 1979*; *Strassberg and DeFelice, 1993*; *Rubinstein, 1995*; *Schneidman et al., 1998*). Taken together, the role of these effects for neural information processing are not yet well understood, and clarification of their exact influence will require experimental access to individual APs across entire axonal arbors.

Mainly due to methodological limitations to track individual APs across multiple sites of mammalian axons, it was not yet possible to record the temporal evolution of a single AP throughout entire axonal arbors. Whole-cell patch-clamp of axonal blebs (*Shu et al., 2006*), boutons (*Alle and Geiger, 2006*), or the Calyx of Held (*Forsythe, 1994*) enables recording of individual axonal signals, however, the technique is invasive and limited to recording from a single site. Axon-attached patch clamp allows for experimental access to intact axons and enables simultaneous recordings from two axonal sites, but stable recording is only possible for less than 60 min and with success rates of ∼50% (*Sasaki et al., 2012*). Owing to the small diameter of neocortical axons (0.08 to 0.4 µm diameter) (*Debanne et al., 2011*), classical electrophysiological tools, such as patch clamp, cannot be used to spatially track the propagation of action potentialat more than a couple sites. Over the past decades, optical microscopy techniques and application of fluorescent indicators, sensitive to voltage (*Peterka et al., 2011*) or calcium (*Grienberger and Konnerth, 2012*), have become popular means for recording electrical activity from neuronal processes in invertebrates (*Stepnoski et al., 1991*; *Fromherz and Müller, 1994*; *Antić and Zecević, 1995*; *Zecević, 1996*; *Antic et al., 2000*) and vertebrates (*Antic et al., 1999*; *Antic, 2003*; *Djurisic et al., 2004*; *Kampa and Stuart, 2006*; *Palmer and Stuart, 2006*; *Canepari and Vogt, 2008*; *Djurisic et al., 2008*; *Acker and Antic, 2009*; *Palmer and Stuart, 2009*; *Holthoff et al., 2010*; *Foust et al., 2010*; *Popovic et al., 2011*; *Popovic, 2015*). Advances in optical electrophysiology techniques (*Popovic, 2015*; *Lin and Schnitzer, 2016*) contributed to accumulating a body of knowledge about neuronal functions (*Popovic, 2015*): voltage-sensitive-dye-based techniques enabled the imaging of individualaction potential propagating into axon collaterals of Purkinje cells, although the cells remained viable for only a few seconds of total illumination (*Popovic et al., 2011*); fluorescence resonance energy transfer (FRET)-based genetically encoded voltage indicators (GEVI) (*St-Pierre et al., 2015*) allowed for tracking of APs with high temporal fidelity (*Chanda et al., 2005*; *Wang et al., 2010*; *Ghitani et al., 2015*); (FRET)-based GEVIs targeted at mammalian axons (*Wang et al., 2012*; *Ma et al., 2017*) enabled to discern axonal APs from the background noise after averaging 10 recording trials (*Ma et al., 2017*). However, optical techniques did not yet provide sufficient temporal resolution and signal quality that would allow for the reliable detection of single individual APs in mammalian axons over longer periods of time.

In the present study, we present a method to noninvasively record individual axonal APs in rat cortical neurons cultured over complementary-metal-oxide-semiconductor (CMOS) –based high-density microelectrode arrays (HD-MEAs) (*Frey et al., 2010*). We use template matching to detect individual axonal APs by combining the simultaneously recorded signals from multiple electrodes. We show recordings of individual APs as they propagate across hundreds-of-micrometer-long-cortical axons at high spatiotemporal resolution. Using the developed method, we measure (I) latencies and temporal precision of AP initiation in response to extracellular voltage stimulation; (II) temporal precision of arrival times of individual APs as they propagate along axonal branches; (III) gradual changes in velocities and temporal precision of propagating APs during high-frequency neuronal activity. We use live imaging to assign the measured extracellular electrical activity to the morphology of individual neurons in the network.

## Results

We recorded cultures of neurons from rat cortex using a state of the art HD-MEA system (*Frey et al., 2010*) (see *Figure 1—figure supplement 1*). Owing to the low-noise recordings provided by our CMOS-based HD-MEAs, averaging over multiple recordings of a single neurons' APs allows for discerning small-amplitude axonal signals from the background noise (*Bakkum et al.,*

*2013*; *Müller et al., 2015*; *Radivojevic et al., 2016*). However, the reliable detection of single axonal APs is needed to investigate the consistency, reliability and precision of axonal conduction, as averaging across multiple APs removes the AP-to-AP variability. Nevertheless, we found that a single axonal AP can be observed if the very same AP is recorded simultaneously on several neighboring electrodes. The detectability of axonal APs can be further improved by combining the signals recorded from multiple electrodes via a matched filtering technique, which we will refer to throughout this work as 'template matching'. In brief (see Materials and methods section for details), the method works by first establishing the prototypical waveform of a neurons' or axons' AP, called its template, on each recording electrode. A multi-electrode filter is then computed from a multi-electrode template that is optimally matched to the individual electrode templates (*Franke et al., 2015*). The filter is then convolved with each (non-averaged) period of recorded data and is analyzed for the presence, or absence, of the neuron's or axon's AP (*Franke et al., 2015*). If the AP is present, the filter will give a large response that is detected by thresholding (see Figure 2a). To estimate the template, we first performed a set of recordings and analytical steps in order to electrically identify axonal arbors of individual neurons within cell cultures of cortical neurons. For this purpose, the high-spatial-resolution of our electrode arrays facilitated 'spike sorting' of extracellular APs which was used for identifying individual neurons and their axons (*Franke et al., 2015*; *Jäckel et al., 2012*). Furthermore, the applied techniques enabled the electrical identification of the respective axon initial segments (AIS), which co-localize with the largest-amplitude and first-emerging extracellular AP traces (*Figure 2—figure supplement 1a*) (*Radivojevic et al., 2016*).

## Spatiotemporal distribution of average extracellular APs captures electrical activity across an entire neuron

For each neuron presented in this study, we used spike-triggered averaging, triggered by the strong extracellular signals recorded close to the neuron's AIS, to capture the neuron-wide AP propagation (*Figure 1a*) (*Müller et al., 2015*; *Radivojevic et al., 2016*). After each spike detected at the AIS, we recorded a 10 ms time window on all electrodes that we refer to as a 'single trial'. As only 126 of the array's 11,011 electrodes can be read out at the same time, for each neuron, we recorded its spontaneous extracellular activity by scanning ~200 different configurations of recording electrodes, in 2 min recordings per configuration, covering the entire array. All ~200 electrode configurations had a small number of electrodes in common. These electrodes were located close to the neurons' AIS and were used for spike detection. For each electrode configuration, we averaged 40 single trials to compute the single-electrode templates. The set of all single-electrode templates provided an array-wide spatiotemporal distribution of the extracellular APs, referred to as an AP 'footprint' or 'electrical image' (*Figure 1a*). The presented footprint spread over 1200 electrodes. Averaging of 40 trials per electrode was needed in order to discern small-amplitude APs of axons and proximal dendrites (*Bakkum et al., 2013*; *Müller et al., 2015*; *Radivojevic et al., 2016*). Reconstructed waveforms of averaged signals recorded in different neuronal regions are presented in the four insets of *Figure 1a*. Neuronal morphology was revealed by lipofection. For details on spike-triggered averaging and lipofection of individual neurons in the network, see the Materials and methods section.

We found a strong dependence between average AP amplitudes on a given electrode and the Euclidean distance of the electrode to the neuron's AIS only within a 100 µm radius. Euclidean distance refers to the geometric distance between the electrode that recorded the largest-amplitude AP waveform and the respective recording electrode. The average decrease in amplitude was ~350 µV at a distance of 100 µm from the AIS. However, we did not find strong decrease in amplitudes over distances between 100 to 1,400 µm from the AIS, showing that distal axonal branches produced similar signals as more proximal branches. Average AP amplitudes varied across an axonal arbor (from ~3 to ~100 µV) independently from the distance of the recording site to the AIS (*Figure 1b*). For the corresponding analysis, we reconstructed AP footprints of 20 neurons (from 18 different preparations) and extracted average AP amplitudes recorded by 26,925 electrodes in total. The spatial profile of the extracellular AP amplitudes is presented in *Figure 1b*. Examples of AP traces and noise recordings are presented in *Figure 2—figure supplement 1a*.

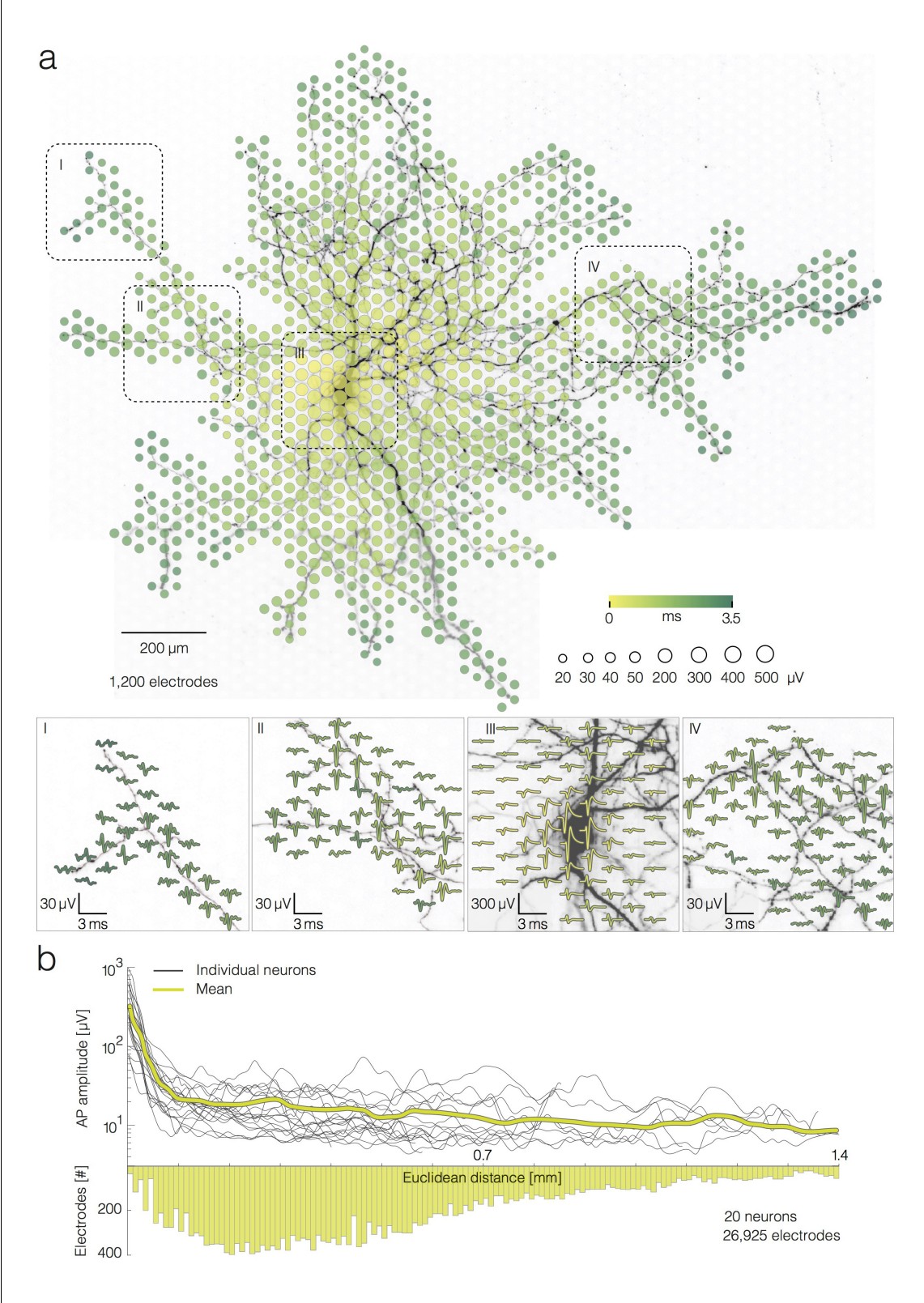

**Figure 1.** Average extracellular APs recorded neuron-wide. (a) Extracellular AP footprint superimposed over neuronal morphology. Circle sizes indicate logarithmically scaled amplitudes of average APs obtained from 40 trials. Arrival time of the APs on different electrodes is color-coded. Close-ups of 4 regions (labeled I, II, III and IV), showing the average AP waveforms (40 trials), are presented at the lower part of (a). Individual APs recorded near the soma and axonal terminal are presented in *Figure 2—figure supplement 1a*. (b) AP amplitudes versus Euclidean distance from the footprint's peak

*Figure 1 continued on next page*

*Figure 1 continued*

signal amplitude location for 20 neurons. Euclidean distance refers to the geometric distance between the electrode that recorded the largest-amplitude AP waveform (presumably at the AIS) and the respective recording electrode, it depends on the inter-electrode pitch. The yellow curve represents mean values of amplitudes over all neurons. The bottom histogram shows the distribution of electrode distances from the electrode that recorded the largest AP amplitude.

DOI: https://doi.org/10.7554/eLife.30198.002

The following figure supplement is available for figure 1:

**Figure supplement 1.** Experimental setup and preparation.

DOI: https://doi.org/10.7554/eLife.30198.003

## Detection of single AP trials across an entire neuron

We used template matching to detect APs within individual trials recorded across the entire neuron. To estimate our capability to detect individual APs in single axonal branches, we used the strong signal on the electrodes close to the AIS as a ground truth of the presence of a given AP. We then tried to detect the same AP independently, only relying on the recording of groups of neighboring electrodes located close to an axonal branch, using several detection strategies: either thresholding individual electrodes or template matching using one, or groups of three or six neighboring electrodes (see *Figure 2* and example in *Figure 2—figure supplement 1b*). In general, we used multiple electrodes to independently detect signals that propagated across an individual axonal branch. This enabled us to discriminate detection errors (i.e., some of the electrodes along a branch do not detect an AP) from possible conduction failures (i.e., none of the electrodes along a branch or axonal segment detects an AP). Template matching strongly improved the detection performance as compared to single electrode thresholding and allowed for detecting a single AP simultaneously and independently in multiple axonal branches with high precision (>90% detection performance). In general, reliable detection of large APs, recorded near the AIS, can be attained by using a simple amplitude threshold set to $5\sigma_{noise}$ (*Quiroga et al., 2004*) (*Figure 2—figure supplement 1a*). However, a simple threshold is inadequate to detect the much smaller APs from axonal arbors, leading to omitted spikes if a high threshold is used, or to false positives due to noise crossing a low threshold.

Performances pertaining to separability of the signal from the noise using template-matching-based detection and simple threshold-based detection are compared in *Figure 2a*. In this example, three neighboring electrodes near an axonal terminal are used to detect axonal spikes. For each electrode, the template matching filter was convolved with the raw data. While the filtering on individual electrodes provided sufficient signal-to-noise ratio for a reliable detection of higher amplitude APs within noisy data (see filter output per electrode in *Figure 2a*), low-amplitude APs buried within the noise might still be missed by the detection, even after template matching. Detection of small-amplitude APs could be facilitated by constructing a multi-electrode filter from templates captured over a local group of neighboring electrodes (*Franke et al., 2015*), which further improved detection performance (see filter output from three electrodes in *Figure 2a*).

We compared performances of the different detection strategies across the entire neuron and found significantly better detectability rates, when template-matching -based detection was used (*Figure 2b*). For the simple threshold-based detection on raw data, recordings from each electrode were treated independently, and the threshold was set to $5\sigma_{noise}$.

To demonstrate the performance of simple threshold-based detection and template-matching -based detection (using inputs from one, three and six neighboring electrodes) over Euclidean distances of recording sites from the soma or AIS, we compared percentages of electrodes that could be used to identify individual APs (*Figure 2c*). Data used for this analysis was obtained from 20 neurons and 23,423 electrodes in total. The grids in *Figure 2—figure supplement 1b* illustrate how electrodes were grouped for the template-matching -based detection.

So far, we only characterized the detection performance in terms of correct detection or misses (false negatives). To assess the fraction of false-positive detections of the template-matching -based technique, trials with APs were mixed with 'noise trials' that have been recorded during inactive neuronal states in a 2:1 ratio (40 AP trials and 20 noise trials per electrode). The false-positives were then expressed as percentage with respect to the total number of detections per electrode (or

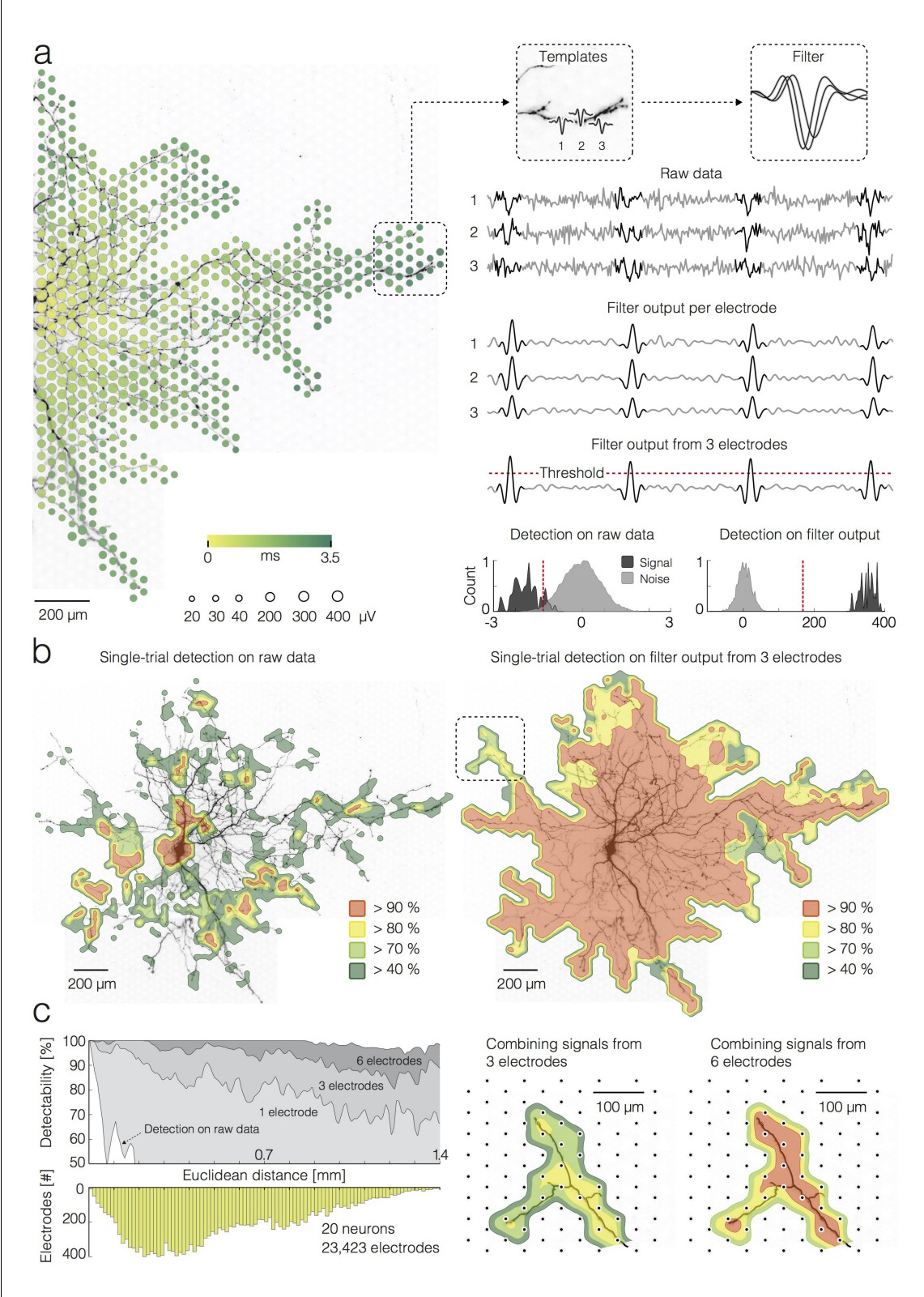

**Figure 2.** Detection of single APs on local groups of electrodes. (a) (left) Extracellular AP footprint superimposed over neuronal morphology. Circle sizes and color code as in *Figure 1(a)*. (right) AP templates recorded near the axonal terminal by three electrodes (denoted as 1, 2 and 3) and the corresponding AP detection filter are framed in boxes. Raw data recorded by the three electrodes (filter input), filter output per individual electrode, and filter output from the group of 3 electrodes are presented as light-gray traces. The AP detection threshold is presented by a red-dashed line. The

*Figure 2 continued on next page*

*Figure 2 continued*

detected signals are marked by dark-gray overlays on the raw data and filter outputs. Separability between signal and noise for the raw data (from the group of 3 electrodes) and for the filter output (from the group of 3 electrodes) are presented in histograms; distributions of signal and noise amplitudes are shaded in dark- and light-gray, respectively; the AP detection threshold is presented by the red-dashed line; the threshold was set to $5\sigma_{noise}$ for simple-threshold detection, and the optimal threshold was computed for template-matching-based detection. (**b**) Detectability of single APs across the neuronal footprint is expressed as percentage of detected signals and presented through contoured surfaces. The obtained detectability is color-coded and superimposed over the neuronal morphology. (left) Signal detection applied on raw data. (right) Signal detection applied on the filter output from groups of 3 neighboring electrodes. (**c**) (left) Efficacy of single-AP detection (percentage of true positives) with respect to the number of electrodes, used to construct a filter (gray shades), and their Euclidean distance from the footprint's peak signal amplitude location. Efficacy of the simple threshold-based detection applied to raw data is presented as a white surface in the graph. The bottom histogram shows the distribution of electrode distances from the electrode that recorded the largest AP waveform. (right) Improvement in AP detectability by increasing the number of electrodes to build the filter from 3 to 6. The close-up region is located in the upper left in *Figure 2b* (see marked box). Euclidean distance refers to the geometric distance between the electrode that recorded the largest-amplitude AP waveform and the respective recording electrode, it depends on the inter-electrode pitch.

DOI: https://doi.org/10.7554/eLife.30198.004

The following figure supplement is available for figure 2:

**Figure supplement 1.** Ground truth, electrode-grouping and false-positive detections.

DOI: https://doi.org/10.7554/eLife.30198.005

group of 3 or six electrodes). We found that false-positive detection was generally less than 5%, whereas an increase in the number of electrodes used for template matching (from 1 to 6) reduced the false-positive detections to ~3% (*Figure 2—figure supplement 1c*). Despite the fact that 40 AP trials were mixed with 20 noise trials, template matching did not yield any false-negative detections in this analysis.

## Precise detection of AP occurrence times

We used template-matching-based detection to extract occurrence times of axonal APs, triggered by extracellular stimulation at the AIS, and we were able to measure neuronal activation latencies caused by different stimulation voltages (*Figure 3*). We selectively stimulated a neuron near the AIS by using positive-first bi-phasic voltage pulses (see Materials and methods section) and we recorded the triggered responses from the axonal arbor of the same neuron (*Figure 3a*). We chose the AIS as a reference stimulation site because, in comparison to other neuronal sites, it requires the lowest voltage for effective and selective activation of individual neurons in the network (*Radivojevic et al., 2016*). A range of voltages (between ±40 and ±80 mV, stepped by ±10 mV) was used to stimulate the neuron. Each voltage was randomly applied 40 times at 1 Hz using the same stimulation electrode. We then recorded stimulation-triggered responses from 25 electrodes placed along a segment of an axonal arbor (*Figure 3a*). Recorded traces (40 trials per stimulation voltage; 200 trials in total) were temporally aligned with respect to the downswing of the stimulation pulses (*Figure 3b*, first column). The alignment between successive triggered AP waveforms then depended on the promptness and on the temporal precision of the neuronal responses to extracellular stimulation (see below). As supra- and sub-threshold voltages were used to stimulate the neuron, the triggered recordings included evoked APs (*Figure 3b*, second column) and stimulation failures, i.e., situations in which no AP was elicited in the neuron (*Figure 3b*, third column).

We were able to extract precise arrival times of the elicited APs with respect to the downswing of respective stimulation pulses at each electrode location. Precise detection of AP arrival times was possible, because the peak of the filter output was found to be a robust estimate of the true temporal position of the AP within the raw data (*Franke et al., 2015*). However, the extracted arrival times represent the result of the summation of two different time spans: (I) the timespan needed for neuronal activation in response to extracellular stimulation (referred to as activation latency) and (II) the timespan needed for an elicited AP to propagate from the stimulation site to the respective recording site (referred to as AP propagation time). To show the activation latencies that have been obtained by applying different stimulation voltages, the APs, triggered by a given voltage, were aligned with respect to the peaks of their filter outputs (*Figure 3b*, fourth column). The temporal shift of the spikes in this alignment procedure reflects its activation latency (larger latencies for smaller stimulation voltages).

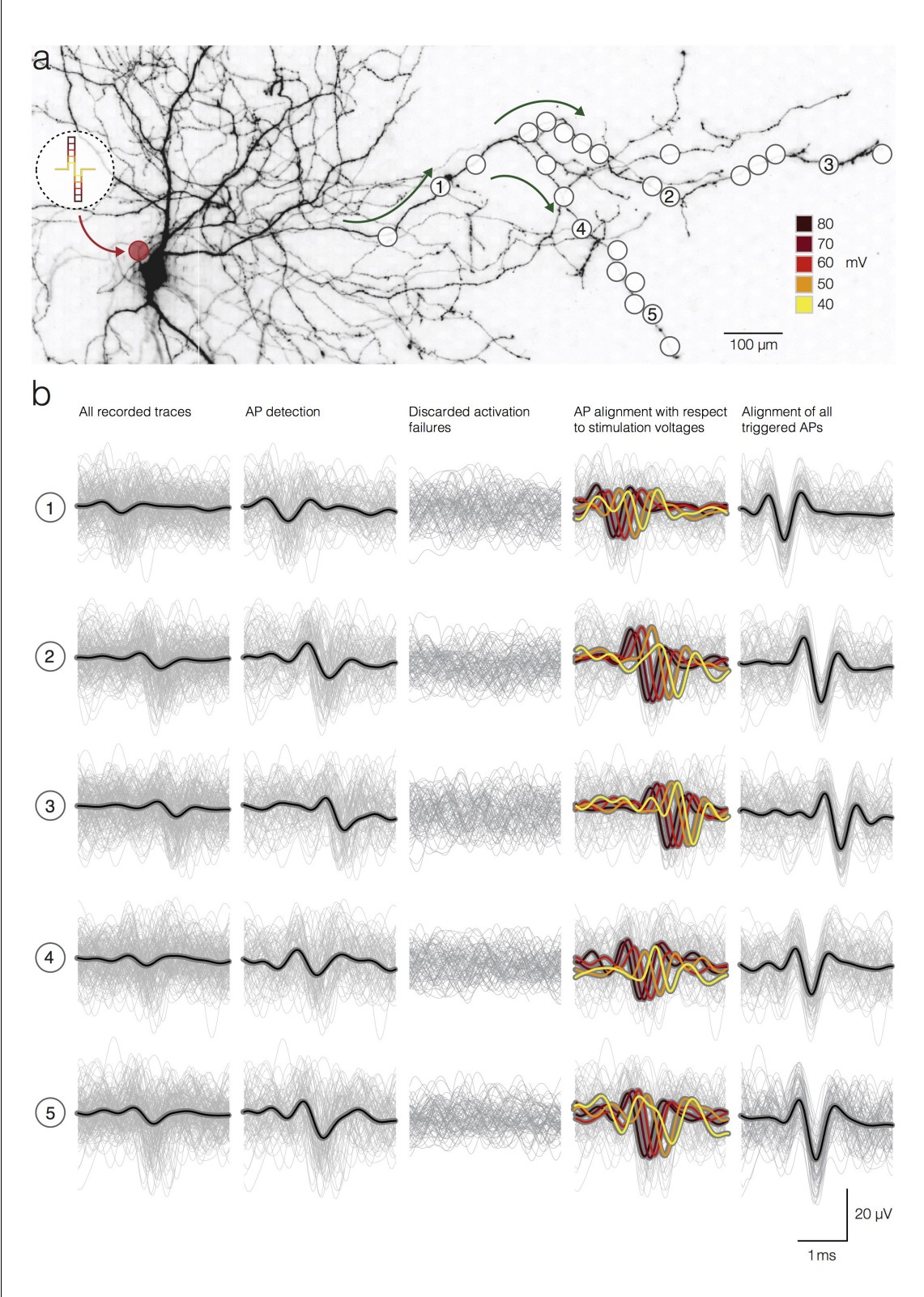

**Figure 3.** Detection and temporal alignment of axonal APs triggered by stimulation. (**a**) Positions of stimulation and recording sites with respect to the neuronal morphology. The red circle near the soma represents the position of the stimulation electrode; white circles mark recording electrodes below axonal branches. Green arrows denote the direction of AP propagation. The biphasic voltage pulses used to stimulate the neuron are depicted within the circular inset. The voltage magnitudes are color-coded. Signals recorded by 5 electrodes (labeled as 1, 2, 3, 4 and 5) are presented below in (**b**). (**b**)

*Figure 3 continued on next page*

*Figure 3 continued*

Stimulation-triggered recorded by the 5 marked electrodes in (**a**) are presented in rows. The columns indicate different stages of the detection process. Average signals are represented by black lines; single trials are represented by gray traces. The first column shows all recorded traces, even those, where no action potential was elicited by the stimulation, temporally aligned on the stimulation onset. The second column shows only the detected APs. The third column shows the discarded activation failures. The fourth column shows detected APs; APs were grouped, and their average waveforms were color-coded according to the magnitudes of the stimulation voltage. The fifth column shows the detected APs aligned according to the time of AP occurrence at the respective electrodes taking into account the respective activation latencies.

DOI: https://doi.org/10.7554/eLife.30198.006

## Increase in the stimulation voltage decreases the latency and increases the precision of neuronal activation

We investigated the effect of stimulation voltage amplitudes on the arrival times of the triggered APs (*Figure 4a–d*) and found that an increase in the stimulation voltage decreases neuronal activation latency (*Figure 4c*) and, at the same time, increases temporal precision of the activation itself (*Figure 4d*). To obtain the data for this analysis, individual neurons were stimulated near their AIS, and the triggered APs were recorded from the proximal part of the neuron's axon (*Figure 4a*). We selected recording electrodes at ~100–500 µm distance from the stimulation site in order to avoid interference of the stimulation artifact with the recorded APs (*Radivojevic et al., 2016*). We used stimulation voltages between ±10 and ±300 mV, with steps of ±10 mV, in order to reconstruct precise excitability profiles across a broad range of stimulation voltages (*Figure 4b*). The different stimulation voltages were applied at 1 Hz in a randomized sequence, using the same stimulation electrode near the AIS. We stimulated 60 times with each voltage. Stimulation-triggered APs were captured by up to three recording electrodes near the proximal part of the axon. In order to obtain good signal quality, we chose electrodes that exhibited comparably large signal-to-noise ratio and used template-matching-based detection to distinguish elicited APs from activation failures. The minimum voltage that triggered an AP in 100% of the stimulation trials was defined as the stimulation threshold (*Figure 4b*). We assigned the obtained excitability profiles to three categories: (I) a sub-threshold regime included stimulation voltages that did not activate the neuron in any of the trials; (II) an intermediary regime included stimulation voltages that were lower than the threshold, but activated the neuron in at least one trial; (III) a supra-threshold regime included stimulation voltages that activated the neuron in 100% of the trials (*Figure 4b*).

To analyze the effect of the stimulation voltage on the neuronal activation latency and activation precision, we first extracted the arrival times of the APs elicited by different voltage magnitudes (*Figure 4a*). The arrival time represents a summation of the activation latency and the propagation time between the stimulation and recording site. Therefore, to calculate activation latencies, we subtracted the estimate of the propagation time from the measured arrival times (see Materials and methods section for details). Using this calculation, we obtained a range of activation latencies for stimulation voltages within intermediate and supra-threshold regimes (*Figure 4c*). We averaged over 60 trials per stimulation voltage (N = 8 neurons from eight different preparations) to obtain AP arrival times and to obtain the estimates of the propagation time.

To analyze the effect of the stimulation voltage on the jitter of the neuronal activation, we first expressed the jitter of the AP arrival time as the standard deviation of arrival times for the respective voltage magnitudes. The jitter of the AP arrival time represents a summation of the neuronal activation jitter and the AP propagation jitter. Therefore, to calculate the jitter of the neuronal activation, we subtracted the estimate of the AP propagation jitter from the jitter of the AP arrival times (see Materials and methods section for details). We used 60 trials per voltage (N = 8 neurons from eight different preparations) to calculate jitters of AP arrival times and to obtain the estimate of the propagation jitter.

In order to demonstrate changes in activation latency and activation precision over different stimulation voltages, we normalized voltages with respect to the intermediate regime of the respective excitability profile. Such normalization was necessary, because excitability profiles obtained from different neurons varied in their intermediate regimes (slopes in (**b**)) and stimulation thresholds (see inset in *Figure 4c*). The difference between the minimum and maximum voltages of the intermediate regime was used as the normalization unit to compare between neurons: for each neuron, we

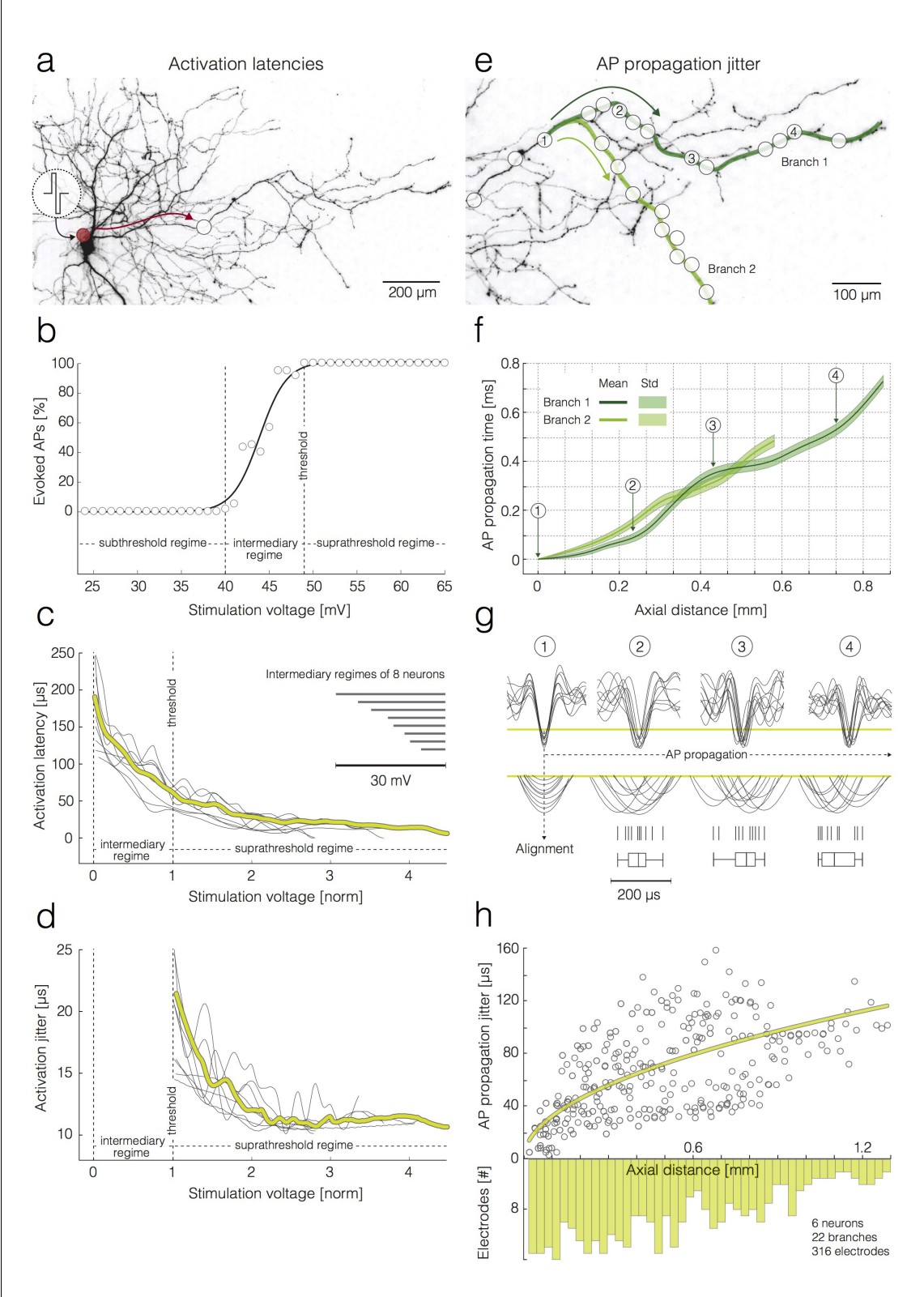

**Figure 4.** Activation latencies and AP propagation jitter. (**a–d**) Dependence of the activation latency and activation precision on the stimulation voltage. (**a**) Experimental design: the red circle denotes the position of the stimulation electrode; the white circle denotes the position of a recording electrode; the red arrow indicates the direction of AP propagation. (**b**) Excitability profile of the neuron shown in (**a**): the stimulation threshold, sub-threshold, intermediary and supra-threshold regimes are indicated in the graph. (**c**) Decrease in activation latency upon increasing the stimulation voltage:

*Figure 4 continued on next page*

*Figure 4 continued*

activation latencies obtained from eight neurons are represented by dark-gray curves; mean activation latency is presented by the yellow curve; the stimulation voltage has been normalized with respect to the stimulation threshold (threshold voltage was set to one); the widths of the intermediary regimes of 8 neurons are presented by dark-gray lines in the upper-right corner. (**d**) Decrease in activation jitter with increasing stimulation voltage; activation jitters obtained from eight neurons are represented by dark-gray curves; the average activation jitter is represented by the yellow curve. (**e–h**) Increase in the AP propagation jitter with axial distance along the axon. Axial distances were measured along the studied axonal branches; they also depend on the inter-electrode pitch. (**e**) Two axonal branches are labeled ('Branch 1' and 'Branch 2') and marked by a dark-green and a light-green line, respectively. White circles indicate the positions of the recording electrodes. (**f**) AP propagation times obtained from the two branches: average propagation times are presented by solid lines; the standard deviations of the propagation times are represented by pale bands in the background. (**g**) APs captured near four axonal sites that correspond to the electrodes labeled as 1–4 in (**e**): the recorded signals were aligned with respect to the AP arrival time detected on electrode 1; the peaks of individual APs (10 APs per electrode) were magnified and projected onto a raster and box-and-whisker plot. (**h**) Increase in the standard deviation (std) of the AP propagation jitter with axial distance from the first detection site near the soma: the AP propagation jitters, obtained from six neurons, are represented by dark-gray circles; the fitted yellow line was obtained by linear regression of the variance of the arrival times against axial distance. The histogram shows the distribution of electrode distances from the location of the first action potential detection.

DOI: https://doi.org/10.7554/eLife.30198.007

normalized the stimulation voltages, $V$, by setting the first voltage to elicit a spike (onset of intermediate regime), $V_{inter}$, to zero, and the threshold (onset of super threshold regime), $V_{thr}$, to one. The normalized voltages were then computed as:

$$V_{norm} = (V - V_{inter}) / (V_{thr} - V_{inter}).$$

Although different neurons had different stimulation voltage ranges associated with the intermediary regime (see inset in *Figure 4c*), their normalized voltage profiles were very similar.

An increase in the stimulation voltage decreased activation latencies (*Figure 4c*; 60 trials per stimulation voltage; N = 8 neurons from eight different preparations). The strongest drop in latencies was within the intermediate regime of the excitability profile (~130 µs drop between minimum and maximum voltage of the intermediate regime). A further increase in the voltage magnitude above the threshold did not drastically decrease activation latencies (~40 µs drop between threshold voltage and supra-threshold voltage at three unit values of the normalized voltage).

An increase in the stimulation voltage beyond the threshold initially increased the precision of neuronal activation (*Figure 4d*; N = 8 neurons from eight different preparations). An increase in the precision by ~12 µs was noticed between threshold voltage and supra-threshold voltage at two unit values of the normalized voltage. A further increase in the voltage magnitude did not drastically increase activation precision. In all stimulation experiments, we used bath application of synaptic blockers in order to avoid possible interference of synaptic inputs with the arrival time estimations of triggered APs.

## Action-potential propagation jitter accumulates along axonal branches

We next analyzed the temporal precision or jitter with which individual APs arrived at given locations along the branch. To compute temporal precision, the arrival time of an AP was detected at a proximal location close to the soma. Then, we estimated the time the AP would take to propagate from the proximal location ('alignment site') along the branch to the next location, where a local electrode group was used to detect it (*Figure 4e–h*). We refer to the standard deviation of this propagation time as the propagation jitter. The propagation jitter of APs increases with the length of the propagation path by 100 µs/mm on average (*Figure 4g,h*). To obtain data for this analysis, we selectively stimulated individual neurons at the AIS while recording triggered APs from multiple axonal branches. Neurons were stimulated at threshold with 100 pulses applied at 1 Hz. Each stimulation pulse was verified to be effective by confirming that an equal number of stimulation pulses and triggered APs was recorded in the axon (*Radivojevic et al., 2016*). Stimulation-triggered APs were captured by 20–120 recording electrodes that were selected along multiple axonal branches of different neurons and chosen for high signal-to-noise ratio SNR (*Figure 4e*).

*Figure 4f* shows propagation times over the length of the propagation path. Because the alignment to the detection at the proximal location canceled out propagation jitter at the alignment site, jitter observed on downstream axonal sites gradually accumulated during propagation along longer

sections of axonal branches (*Figure 4g,h*). Propagation jitter obtained from 22 branches of 6 neurons (from six different preparations) are shown in *Figure 4h*.

## High-frequency neuronal activity decreases velocity and increases jitter of AP propagation

We also investigated propagation times of single APs during high-frequency activation and found a decrease in velocity and an increase in the jitter of AP propagation (*Figure 5*). To obtain data for this analysis, we selectively stimulated individual neurons at the AIS while recording triggered APs from the axonal arbor (*Figure 5a*). Neurons (N = 5 neurons from five different preparations) were repetitively stimulated at 100 Hz, using threshold voltages. Stimulation protocols were organized in 100 stimulation episodes, whereas, each episode was composed of 100 stimuli applied at 100 Hz. Therefore, we obtained 100 single trials (from 1st to 100th) per each stimulation episode. An interval of 60 s separated consecutive stimulation episodes. Electrical stimulation near the AIS was effective for each stimulation for the given protocol, which was confirmed by observing an equal number of stimulations and triggered AP events recorded from the axonal arbor (*Radivojevic et al., 2016*). Stimulation-triggered APs were captured by 20–120 recording electrodes selected along multiple axonal branches, which exhibited high SNR.

To investigate whether repetitive neuronal activation at 100 Hz affects propagation velocities and jitter, triggered APs were grouped with respect to the stimulation count (from 1st to 100th). We used template-matching-based detection to extract the arrival times of individual AP across all stimulations. Arrival times were measured with respect to AP arrival times recorded from a proximal axonal site (alignment site, indicated as electrode one in *Figure 5a*), which enabled us to compare AP waveforms (*Figure 5a*), propagation times (*Figure 5b*) and propagation jitters (*Figure 5c*) across different stimulation trials (from 1st to 100th).

Repetitive stimulation at 100 Hz decelerated AP propagation gradually across 100 successive stimulations (*Figure 5b*). 100 Hz stimulation also increased AP propagation jitter across successive stimulations (*Figure 5c*). Axial distances were measured along the specific observed axonal branch with all its turns; axial distances are also depending on the inter-electrode pitch (17.8 μm center-to-center pitch). Data presented in *Figure 5b–d* and *Figure 2—figure supplement 1a,b* were obtained from the neuron shown in *Figure 5a*.

The same analysis was repeated on 12 branches of 5 neurons from five different preparations (*Figure 5e,f*). We used linear regression to obtain propagation velocities and propagation jitter of APs triggered by the 1st, 25th, 50th and 100th stimulation. Repetitive neuronal activation at 100 Hz induced a decrease in AP propagation velocity by 20.7% (0.14 ± 0.2 m/s (mean ± SEM), p<0.001, paired t-test, N = 12) between the 1st and 100th stimulation event (*Figure 5e*). Estimated velocities were 0.71 ± 0.07, 0.65 ± 0.06, 0.62 ± 0.06 and 0.57 ± 0.05 m/s (mean ± SEM) for 1st, 25th, 50th and 100th stimulation event, respectively (inset in *Figure 5e*). Propagation jitter increased by 11.7% or 11.7 ± 5.24 μs/mm (mean ± SEM, p=0.03, paired t-test, N = 12) between the 1st and 100th stimulation event (*Figure 5f*). The estimated propagation jitter was 111 ± 4.32, 115 ± 4.55, 121 ± 3.82 and 124 ± 8.34 μs/mm for the 1st, 25th, 50th and 100th stimulation event, respectively (inset in *Figure 5f*).

## Discussion

We cultured neurons taken from whole primary neocortices of rats over HD-MEAs and developed a method to noninvasively detect single individual APs propagating across hundreds-of-micrometer-long cortical axons. Live imaging was used to correlate extracellular electrical activity to the morphology of individual neurons in the network.

Spike-triggered averaging of extracellular APs, triggered by signals recorded near the AIS, enabled the reconstruction of AP footprints - a spatiotemporal representation of the average electrical activity recorded across an entire neuron (*Figure 1a*). This was possible owing to: (I) electrical identification of the AIS based on the extracellular activity pattern of the neuron (*Bakkum et al., 2013*; *Radivojevic et al., 2016*); (II) reliable detection of the comparably large AIS signals that served as a trigger for the averaging (*Figure 2—figure supplement 1a*), and (III) low-noise recording provided by the array (*Frey et al., 2010*), which allowed to discern small-amplitude APs of the neurites after averaging (*Bakkum et al., 2013*; *Radivojevic et al., 2016*). In general, an AP footprint

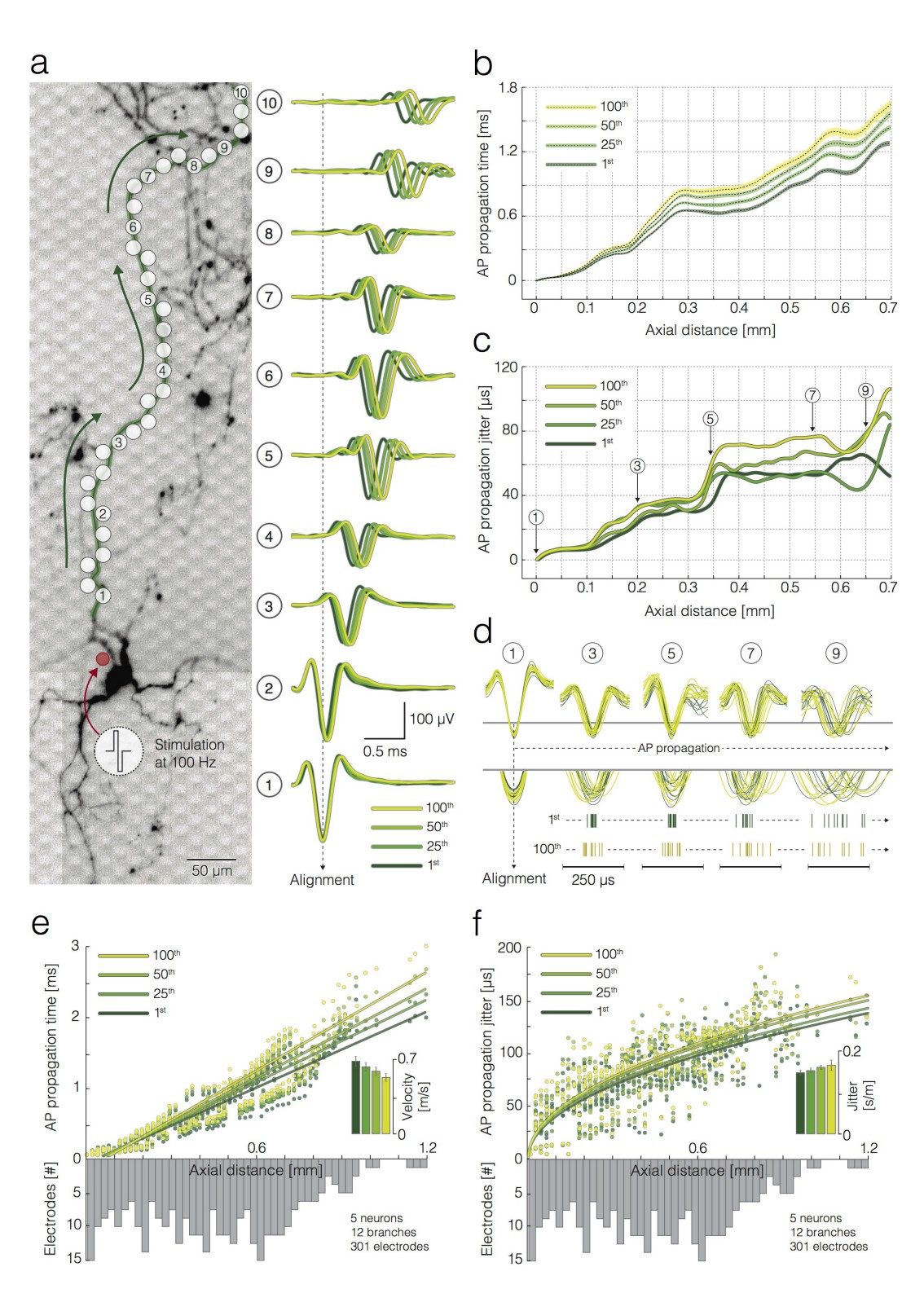

**Figure 5.** Activation-frequency-dependent changes in AP propagation times and jitters. (a) (left) Position of stimulation and recording sites with respect to the neuronal morphology. The red circle represents the stimulation electrode; the white circles mark recording electrodes along an axon. Biphasic voltage pulses, used to stimulate the neuron, are depicted within the circular inset. The neuron was stimulated 100 times with 100 pulses at 100 Hz. Signals recorded by 10 electrodes (labeled as 1–10) are presented at the right. (right) Average APs recorded by the labeled electrodes were aligned

*Figure 5 continued on next page*

*Figure 5 continued*

with respect to the arrival time of the signal detected on electrode 1. Average APs, triggered by the 1st, 25th, 50th and 100th stimulation pulse, are color coded. (b) Propagation times of APs triggered by the 1st, 25th, 50th and 100th stimulation: average propagation times are presented as black dashed lines; standard errors (SEM) are color-coded and represented by pale bands. The data were obtained from the neuron shown in (a). (c) AP propagation jitters, observed during neuronal response to the 1st, 25th, 50th and 100th stimulation, are represented through color-coded curves. The data were obtained from the neuron shown in (a). (d) Individual AP trials, triggered by the 1st and 100th stimulation are colored in green and yellow; the APs were recorded near 5 axonal sites (labeled as 1, 3, 5, 7 and 9) that correspond to the sites marked in (a, c); the recorded signals were aligned with respect to the arrival time of APs detected on electrode 1; aligned signals, triggered by the 1st and 100th stimulation pulse, were superimposed and centered with respect to occurrence time for the sake of comparison; the peaks of individual APs' were magnified and projected onto raster plots. Note the differences in the jitter of AP trials triggered by the 1st and 100th stimulation pulse. (e) Deceleration of AP propagation during high-frequency stimulation: the average AP propagation times, observed during neuronal response to the 1st, 25th, 50th and 100th stimulation pulse, are presented through color-coded circles; the data obtained from four neurons (12 axonal branches in total) are jointly displayed; fitted lines were obtained by using linear regression. Inset: the color-coded histogram shows AP propagation velocities during neuronal response to the 1st, 25th, 50th and 100th stimulation pulse; the values were calculated from the slope of the linear regression. (f) Increase in AP propagation jitter during high-frequency stimulation: the AP propagation jitter, observed during neuronal response to the 1st, 25th, 50th and 100th stimulation, are presented through color-coded circles; the data obtained from four neurons (12 axonal branches in total) are jointly displayed; the fitted lines were obtained using linear regression of the variance of the arrival times against axial distance. Inset: the color-coded histogram shows the propagation jitter observed during neuronal response to the 1st, 25th, 50th and 100th stimulation; the values were calculated from the square root of the slope of the linear regression. Axial distances in (b), (c), (e) and (f) were measured along the specific observed axonal branch with all its turns; they are also depending on the electrode pitch (17.8 µm center-to-center pitch).

DOI: https://doi.org/10.7554/eLife.30198.008

captures electrical activity across the AIS, axonal trunk and lower-order branches, which enables to electrically visualize large portions of the axonal arbor and to track AP propagation in multiple axonal branches (*Bakkum et al., 2013*; *Müller et al., 2015*; *Radivojevic et al., 2016*). The soma and proximal dendrites provide minor contributions to the extracellular footprint, which are typically masked by the much larger AIS signals (*Radivojevic et al., 2016*). Owing to their small extracellular signals, distal dendrites seem not to be detectable with the HD-MEA devices used here (*Bakkum et al., 2013*).

We analyzed AP footprints obtained from 20 neurons and observed a strong decrease in the average amplitude of the extracellular APs over a 100 µm Euclidean distance from the AIS (*Figure 1b*). This can be explained by the electrical contribution of voltage-gated ion channels, whose concentration is high at the AIS and significantly drops moving towards somatodendritic and axonal compartments (*Debanne et al., 2011*).

We found that AP amplitudes broadly varied across neurites, however, such variation was largely independent of the Euclidean distance of the recording site to the AIS (*Figure 1b*). The amplitude of an extracellular AP is proportional to the sum of all transmembrane currents (*Gold et al., 2007*) weighted by the relative proximity of the current sources to the recording electrode, the filter properties of medium and electrode, and the characteristics of the analog filters in the recording circuitry and of the digital filters used for data analysis. Whereas the filter properties can be safely assumed to be identical for all electrodes (*Obien et al., 2014*), the distance between the membrane and electrode likely varies across axonal arbors and recording sites. Therefore, axonal compartments that are tightly attached to the electrode surface could provide large signal amplitudes, whereas axonal sections that are between electrodes could render much smaller amplitudes. In addition, the electrode-neuron interface could be affected by local interactions between axons and glial cells. Thus, for example, glial cells may elevate axons away from the array's surface, making them less accessible to recording. Alternatively, they might cover an axon over an electrode, and thereby tighten the electrode-axon contact and increase signal heights. Supporting evidence from our previous study shows that axonal sensitivity to extracellular stimulation also varies across an axonal arbor, which might be caused by similar factors (*Radivojevic et al., 2016*). Besides the quality of the electrode-neuron contact, the biophysical properties of an axon itself could explain amplitude variations along the arbor. Differences in the distributions of voltage-gated ion channels and various morphological features, such as branching points and axonal varicosities, could, for example, play a role in determining AP amplitudes (*Debanne et al., 2011*).

In general, APs recorded near the AIS were, on average, approximately 10 times larger in their amplitudes compared to those captured near axonal arbors and dendrites (*Figure 1b*; *Figure 2—*

*figure supplement 1a*). The extracellular amplitude depends on the transmembrane currents of each neural compartment, but also on the spatial arrangement of the compartment with respect to the specific electrode that is recording the signal. This means that one cannot directly derive quantitative information about the transmembrane currents from the extracellular recording. The high density of electrodes and the availability of a multitude of electrodes at different distances from a signal-generating source or neuronal compartment, however, do enable to derive a spatial profile of signal intensity. The high amplitudes at the AIS signals could be reliably detected within a single trial using simple threshold-detection (*Figure 2—figure supplement 1*). Reliable detection of signals from the AIS can then be used to facilitate detection of small-amplitude signals recorded from the axonal arbor. Under the assumption that AP propagation velocity is, within certain boundaries, constant over subsequent trials (*Bakkum et al., 2013*; *Bakkum et al., 2008*), it can be expected that individual electrodes near distal axons always captured APs at approximately the same time with respect to the time of the AIS signal. In this sense, the large-amplitude signal from the AIS could serve as a ground truth for the detection of small-amplitude signals (*Figure 2—figure supplement 1a*), however, it did not allow for discerning individual APs within the background noise. Therefore, averaging over 40 trials was previously needed to discern small-amplitude signals recorded from the axonal arbor (*Bakkum et al., 2013*; *Müller et al., 2015*; *Radivojevic et al., 2016*).

By using template matching with multi-electrode templates, we were now able to detect individual APs propagating across an entire neuron (*Figure 2b*), including APs recorded near axonal terminals hundreds of micrometers away from the AIS (*Figure 2a*). This was possible using the optimal matched filter, a finite impulse response filter, which maximizes the signal-to-noise ratio of a given transient signal in the background noise when convolved with the recordings (*Franke et al., 2015*). Consequently, separability of the signal from the noise obtained through template-matching-based detection was significantly better compared to that provided by a simple threshold-detection (*Figure 2a*). Both, detection failures and false-positive detections could be strongly reduced by constructing a multi-electrode filter using signals recorded by a group of three or six neighboring electrodes (*Figure 2*; *Figure 2—figure supplement 1b,c*).

We found that an increase in stimulation voltage decreased activation latency (*Figure 4c*) and increased precision of the activation itself (*Figure 4d*). A decrease in the latency could be explained by the fact that more charge activates a higher number of ion channels at once, leading to faster recruitment of other channels in the membrane and eventually to faster activation of the AIS.

In our measurements, AP propagation jitter accumulated during axonal conduction (*Figure 4g,h*). We determined the arrival time jitter to be 100 µs per mm axonal length. Arrival time jitter of individual APs will impact the reliability of coincidence detection of multiple PSPs at the postsynaptic neuron and will, therefore, impact neural-network computations based on a temporal code (*König et al., 1996*). However, for unmyelinated axons of less than a few mm length, other sources of variability, such as temporal imprecision of synaptic transmission, synaptic transmission failure, or PSP amplitude variability, seem to contribute to a larger extent to imprecision in information transmission (*Ribrault et al., 2011*; *Krächan et al., 2017*).

The inconsistency in AP propagation time could be attributed to the thermodynamic noise inherent to the gating dynamics of voltage-gated ion channels in axons (*Hodgkin and Huxley, 1952*; *White et al., 2000*; *DeFelice, 2012*). Supporting evidence from modeling studies suggests that such noise can cause threshold fluctuations in axonal membrane and affect reliability of AP initiation in membrane patches (*Skaugen and Walløe, 1979*; *Strassberg and DeFelice, 1993*; *Rubinstein, 1995*; *Schneidman et al., 1998*). Considering theoretical studies, the accumulation of AP jitter along a conduction path may be influenced by two factors: (I) the cumulative afflux of the noise along the propagation path, and (II) more noise being introduced, as the axonal diameter decreases from proximal regions towards axonal terminals (*Faisal et al., 2005*). Other phenomena, such as ephaptic coupling and the presence of gap-junctions could also be a source of electrical noise in unmyelinated axons, although we did not observe any related evidence. Furthermore, axonal morphology is not uniform along the arbor, and it is possible that branching points and axonal varicosities locally impact on AP propagation. We also noticed that cortical axons conducted single action potentials with high reliability: in more than 8,000,000 recorded action potentials we did not observe any conduction or branch-point failure.

In a next set of experiments, we found that the high-frequency regime of neuronal activity induced a decrease in velocity and an increase in jitter of the AP propagation (*Figure 5*). Previous

studies have reported deceleration of AP propagation velocities during repetitive electrical stimulation (*Soleng et al., 2003*; *Raastad and Shepherd, 2003*; *Shimba et al., 2015*), but, to the best of our knowledge, this is the first time that the propagation jitter, accumulated during AP conduction, and its increase by high firing rates were measured. Repetitive neuronal activation at high frequency leads to an increase in the intracellular $Na^+$ concentration (*De Col et al., 2008*), which hyperpolarizes the axonal membrane through activation of $Na^+/K^+$ ATPase (*Soleng et al., 2003*). Such hyperpolarization could explain the deceleration of AP propagation during high-frequency stimulation (*Soleng et al., 2003*; *Shimba et al., 2015*). Alternatively, accumulation of extracellular $K^+$ could contribute to $Na^+$ channel inactivation by keeping the membrane depolarized for a longer time after a stimulus (*Malenka et al., 1981*; *Poolos et al., 1987*), the decrease in AP propagation velocities could then be attributed to more time needed for $Na^+$ channels to recover from inactivation.

## Materials and methods

### Animal use

All experimental protocols were approved by the Basel Stadt veterinary office according to Swiss federal laws on animal welfare and were carried out in accordance with approved guidelines (*Tierschutzgesetz* TSchG, SR 455; *Tierschutzverordnung* TSchV, SR 455.1).

### HD-MEAs

A complementary-metal-oxide-semiconductor (CMOS)-based HD-MEA system, fabricated in a 0.6 µm CMOS 3M2P process was used for extracellular neuronal recording and stimulation (*Frey et al., 2010*) (see *Figure 1—figure supplement 1*). The electrode array has been co-integrated with circuitry units on the same chip and featured a total of 11,011 electrodes (active surface of $8.2 \times 5.8$ µm$^2$ per electrode) within an area of $1.99 \times 1.75$ mm$^2$, providing a density of 3150 electrodes per mm$^2$ (17.8 µm center-to-center pitch). The chip surface has been passivated with a stack of alternating $SiO_2$ and $Si_3N_4$ layers. Bond-wires were encapsulated in epoxy (Epo-Tek 302–3M, John P. Kummer AG, Cham, Switzerland). Owing to a flexible switch matrix and 13,000 static random-access memory cells integrated underneath the array, up to 126 readout and/or stimulation channels could be routed to the desired electrodes and reconfigured within a few milliseconds. Electrodes exhibited signal-to-noise ratios up to $180\sigma_{noise}$ for somatic and between 1 and $20\sigma_{noise}$ for axonal signals, respectively. On-chip circuitry was used to amplify (0–80 dB programmable gain), filter (high pass: 0.3–100 Hz, low pass: 3.5–14 kHz), and digitize (8 bit, 20 kHz) the neuronal signals. Digitized signals were sent to a field-programmable gate array (FPGA) board and further streamed to a host PC for real-time visualization and data storage. Recorded signals were up-sampled to 200 kHz following the Whitaker–Shannon interpolation formula. Matlab R2012a was used for data analysis and to design extracellular stimulation protocols.

### Platinum-black deposition

We deposited platinum black on the electrodes in order to reduce their impedance and to increase the effective electrode-neuron interfacing area. Consequently, stimulation voltages, needed to elicit neuronal responses, were reduced, which significantly improved stimulation performance of the array. A 180 mA current was simultaneously applied to all electrodes for 45–75 s while using a platinum counter electrode immersed in the deposition solution (0.7 mM hexachloroplatinic acid and 0.3 mM lead (II) acetate anhydrous). Deposition uniformity was verified optically under a microscope and electrically by measuring the impedance of the electrodes (see below).

### Impedance measurement

To verify that all electrodes provide even current flow, we measured their impedances in response to a test voltage signal applied at each electrode. We found negligible variations in impedance (<1%) across all electrodes on 8 chips. All measurements were performed in phosphate-buffered saline (PBS; Sigma, Buchs, Switzerland) by using an external DS360 ultra-low distortion function generator (Stanford Research Systems; Sunnyvale, CA, U.S.) to issue voltage pulses and the integrated amplifiers on the HD-MEA system for readout of the respective signals. The external function generator sent a sinusoidal voltage signal ($V_{stim}$; 1mV$_{pp}$ at 1 kHz) through the chip's reference electrode

into the PBS medium, whereas active electrodes were used to record the resulting signal through on-chip readout channels. The attenuation of the recorded signal was dependent on the impedance of the active electrode ($Z_e$) and that of the amplifier input ($Z_{in}$). The amplifier inputs of acquired signals were I-Q demodulated in Matlab R2012a in order to extract the amplitude of the input reference signal ($V_{meas}$) at 1 kHz.

## Cortical cultures

We used a culturing protocol developed for long-term maintenance of neural cultures (*Lin and Schnitzer, 2016*). We introduced minor adaptations to the protocol in order to constrain culture growth to the area of the array and to maintain optimal conditions of growth media during long-term experimentation. Cortices from embryonic day 18 Wistar rat (RRID:RGD_2308816) were dissociated enzymatically in trypsin with 0.25% EDTA (Thermo Fisher Scientific, Bleiswijk, Netherlands) and physically by trituration. For cell adhesion, a layer of 0.05% polyethyleneimine (Sigma) in borate buffer (Chemie Brunschwig), followed by a layer of 0.02 mg ml$^{-1}$ laminin (Sigma) in Neurobasal (Thermo Fisher Scientific) was deposited on the electrode array. To constrain culture growth to the electrode array, a cell-drop containing ~20,000 cells and covering ~3 mm$^2$ was seeded in the center of the array. The plating media were changed to growth media after 6 days. Plating media consisted of Neurobasal, supplemented with 10% horse serum (HyClone), 0.5 mM GlutaMAX and 2% B27 (Thermo Fisher Scientific). Growth media consisted of DMEM (Thermo Fisher Scientific), supplemented with 10% horse serum, 0.5 mM GlutaMAX and 1 mM sodium pyruvate (Thermo Fisher Scientific). Cultures were maintained inside an incubator under controlled environmental conditions (36°C and 5% $CO_2$). Experiments were conducted at 14–28 DIV. The culturing chambers were sealed with a ~ 1 mm layer of light mineral oil (Sigma) floating above the growth medium. The sealing provided selective permeability to gases, such as $O_2$ and $CO_2$, and prevented evaporation and consequent changes in growth media's osmolarity during long-term experiments. To block synaptic inputs, we inhibited GABA-R, NMDA-R and AMPA-R by bath application of 50 μM bicuculline methiodide, 100 mM 2-amino-5-phosphonovaleric acid and 10 mM 6-cyano-7-nitroquinoxaline-2, 3-dione (CNQX; Sigma), dissolved in growth media.

## Immunocytochemistry

Cortical cultures were fixed in 4% paraformaldehyde (Thermo Fisher Scientific) in PBS (Sigma) at pH 7.4 for 15 min at room temperature, washed twice with ice-cold PBS, permeabilized with 0.25% Triton X-100 (Sigma) in PBS for 10 min and washed three times in PBS. Fixed cultures were exposed to phosphate-buffered saline with tween 20 (1% bovine serum albumin and 0.1% tween 20 in PBS; Sigma) for 30 min to prevent unspecific binding of antibodies. The primary antibodies Anti-beta III Tubulin antibody [TU-20] (Abcam Cat# ab7751, RRID:AB_306045), diluted in phosphate buffered saline with tween 20 to a ratio of 1:500, were added and left overnight at 4°C on a shaker. Cultures were washed three times in PBS for 5 min each time on the shaker. The secondary antibodies Alexa Fluor 647 (Thermo Fisher Scientific Cat# A-21449, RRID:AB_2535866), diluted to ratio of 1:200 in PBS with 1% BSA, were added and left for 1 hr in the dark at room temperature. Samples were washed three times in PBS for 5 min in the dark and then stored at 4°C.

## Live imaging and microscopy

Live-cell visualization of whole neurons was performed by transfection using pLV-hSyn-RFP plasmid from Edward Callaway (Addgene plasmid # 22909) and Lipofectamine 2000 (Thermo Fisher Scientific) in accordance with the manufacturer's protocol. Transfections were carried out in serum-reduced OptiMEM media (Thermo Fisher Scientific). A Leica DM6000 FS microscope, Leica DFC 345 FX camera, and the Leica Application Suite software were used to produce micrographs. We performed live-cell visualization of 2 neurons from two different preparations in order to illustrate our experimental approach (*Figures 3a*, *4a,e* and *5a* and *Figure 2—figure supplement 1a,b*) as well as to demonstrate that the spatiotemporal distribution of the recorded signals matches with neuronal morphology (*Figures 1a*, *2*, *3a*, *4e* and *5a* and *Figure 2—figure supplement 1a,c*). The latter has also been demonstrated in previous studies, where live-cell visualization was used to verify axonal AP propagation tracking by means of HD-MEA-based monitoring of extracellular APs (*Bakkum et al., 2013*; *Radivojevic et al., 2016*).

## Electrical identification of individual neurons in the network

To electrically identify individual neurons in the network, we used approaches thoroughly described in our previous study (*Radivojevic et al., 2016*). To initially identify the locations of neurons, array-wide spontaneous activity of the network was captured by sequential scanning of 95 recording configurations. In each configuration, up to 126 randomly selected electrodes sampled neuronal activity for a duration of 60 s. The average voltage traces, recorded by each electrode, were extracted and used to reconstruct the map of the network's electrical activity. Since the largest extracellular signals occur near the AIS (*Radivojevic et al., 2016*), and signals from axonal arbors have much smaller amplitude, regions with large-amplitude signals in the activity maps indicated the locations of the AISs. Spike-triggered averaging, synchronized with the putative AIS spikes, enabled us to reconstruct the spatiotemporal distribution of extracellular AP waveforms arising from neuronal processes (*Figure 1a*). The first step in obtaining these data was selecting electrodes with the four largest spike amplitudes per putative AIS region found in the activity map. We next designed multiple recording configurations covering the entire array, where, in each configuration, 4 of the 126 read-out channels were set as the four preselected electrodes. Other available recording channels were connected to randomly selected electrodes. Each configuration was used to sample neuronal activity during 2 min. Spikes recorded by the four preselected electrodes in each configuration were sorted by using the UltraMegaSort2000 software (*Hill et al., 2011*), and the timestamps of the neuron's initial spike were extracted. Array-wide spike-triggered average signals were computed, and their spatiotemporal distribution (also referred to as AP footprint) was reconstructed by using a custom designed Matlab code.

## Selective stimulation of individual neurons in the network

In all stimulation protocols, we used balanced positive-first biphasic voltage pulses, with phase durations of 200 μs and amplitudes between ±10 and ±300 mV, applied to one electrode at a time. Elicited neuronal responses were estimated by observing stimulation-triggered APs recorded from pre-identified axonal branches (see above). To ensure selective and reliable activation of individual neurons in the network, we used approaches thoroughly described in our previous study (*Radivojevic et al., 2016*). In brief, we applied neuron-wide stimulation over a range of voltages and revealed that sites with the lowest activation threshold resided within a proximal part of the neuron's extracellular AP footprint, near the largest and first-emerging AP voltage traces. Stimulation was applied at 4 Hz for voltages from ±10 to ±300 mV, with steps of ±10 mV. Each stimulation voltage was applied 60 times per site, and thresholds were defined as the minimum voltage to trigger an AP in 100% of the trials. To get more precise excitability profiles, the most sensitive sites were then stimulated with voltages stepped by 1 mV (60 stimuli per step in random order). To next ensure that high-frequency stimulation near the most sensitive sites provided reliable neuronal responses, we stimulated the neuron 100 times with a series of 100 pulses streamed at 100 Hz, and chose the stimulation electrode that provided neuronal activation in 100% of the trials at the lowest voltage. To further verify whether the chosen stimulation parameters provided selective activation of a single neuron, instead of activating other neurons in the culture, we observed stimulation-triggered responses across the entire array and checked that no other large-amplitude APs (i.e. AIS spikes) were evoked. In all cases presented here (8 neurons in eight different preparations), threshold stimulation near the largest and first-emerging trace of the neuron's AP footprint provided selective and reliable neuronal activation at comparably low stimulation voltages of around 50 mV. In all stimulation experiments, bath application of synaptic blockers was used to avoid possible interference from synaptic inputs when measuring neuronal excitability and AP propagation.

### Template-matching-based AP detection

An extracellular footprint was obtained through spike-triggered averaging triggered at the AIS. This was possible, as the AIS produced electrical signal amplitudes large enough to be reliably detected through a simple amplitude thresholding. The extracellular footprint was used as a template for the detection of single directly-evoked APs that propagated across an entire neuron (*Figure 1a*). For this purpose, sets of electrodes across an entire footprint were grouped into local clusters that contained 3–6 electrodes in close proximity (*Figure 2—figure supplement 1b*). Single trials recorded by each electrode were next convolved with a matched filter, derived from the template at that

electrode (*Franke et al., 2015*), and the result was summed over the electrode cluster. The result of this convolution was the template matching output. When the template matching output crossed a threshold, the presence of a spike was detected. As the peak of the template matching output is a robust estimate of the true temporal position of the AP within the raw data (*Franke et al., 2015*), we were able to extract precise occurrence times of individual APs (*Figure 3*). This was particularly important for extracting occurrence times of small-amplitude APs with very low signal-to-noise ratio. To compute the matched filter and the optimal threshold to detect spikes, the noise statistics (*Pouzat et al., 2002*) have to be estimated. We used periods of the data, where no spikes could be detected in the raw recordings, to compute the standard deviation of the noise and the noise covariance matrix. The template matching procedure used here provided an optimal detection threshold analytically and did not require manual intervention (*Franke et al., 2015*).

### Propagation time and activation latency

The arrival time of an AP ($T_{arrival}$), elicited by extracellular stimulation at the AIS, and recorded from the proximal axon, represents a summation of the activation latency ($T_{activation}$) and the propagation time ($T_{propagation}$) between the stimulation and recording site ($T_{arrival} = T_{propagation} + T_{activation}$). To estimate the propagation time, we assumed that the activation time is close to zero for high stimulation voltages ($V_{suprathreshold}$), as those were found to provide immediate activation ($T_{immediate\_activation}$) when applied to the neuron's AIS (*Radivojevic et al., 2016*) (if $V_{stimulation} = V_{suprathreshold} \Rightarrow T_{activation} = T_{immediate\_activation} \approx 0$). Therefore, the propagation time corresponded to the respective arrival time for the largest stimulation voltage ($T_{propagation} = T_{arrival} - T_{immediate\_activation}$). To calculate activation latencies for lower stimulation voltages, we subtracted the estimate of the average propagation time from the measured arrival times ($T_{activation} = T_{arrival} - T_{propagation}$). We averaged over 60 trials per stimulation voltage (N = 8 neurons from eight different preparations) to express average activation latencies and to obtain the estimate of the average propagation time.

### Activation jitter

The jitter of the AP arrival time ($J_{arrival}$) was expressed as a standard deviation of AP arrival times ($J_{arrival} = std(T_{arrival})$) and represents a summation of the activation jitter ($J_{activation}$) and the propagation jitter ($J_{propagation}$; $J_{arrival} = J_{activation} + J_{propagation}$). To estimate the effect of the stimulation voltage on the activation jitter, we assumed that the AP propagation jitter does not depend on the stimulation voltages ($J_{propagation} = const.$). Therefore, the decrease in the jitter of AP arrival time with increasing stimulation voltages could be attributed exclusively to a decrease in the activation jitter ($J_{arrival} = J_{activation} + const.$). The lowest measured jitter of AP arrival times, measured when neurons were stimulated with high voltages, were used as the estimate of the AP propagation jitter. Therefore, we were able to calculate the activation jitter by subtracting the estimate of the propagation jitter from jitters of AP arrival times for different voltages ($J_{activation} = J_{arrival} - J_{propagation}$). We used 60 trials per voltage (N = 8 neurons from eight different preparations) to calculate jitters of AP arrival times and to obtain the estimate of the propagation jitter. The distance between stimulation and recording electrodes was <500 μm, and the obtained propagation jitters were <50 μs.

### Propagation jitter

We computed the propagation jitter at a single electrode as the standard deviation of the arrival times of individual action potentials at that electrode. To estimate the propagation jitter as a function of distance from the reference electrode (where the jitter was set equal to zero), we used linear regression: If the jitter introduced by each axonal segment was independent of that introduced by the other segments, the measured jitter over axial distance could be described as a white noise Gaussian process, so that variance increased linearly with axial distance. Therefore, we fitted the variances at each electrode as a function of axial distance by a linear function with a y-intercept of zero. The square root of the slope of this regression then yielded the average propagation jitter per unit length of axon.

## Acknowledgements

This work was supported by the European Community through the European Research Council Advanced Grants 267351'NeuroCMOS' and 694829 'neuroXscales', as well as the Swiss National

Science Foundation through Grant 205321_157092/1, the Ambizione Grant PZ00P3_132245 and the Ambizione Grant PZ00P3_167989. The funders had no role in study design, data collection and analysis, decision to publish, or preparation of the manuscript. We thank Alexander Stettler and Albert Martel for post-processing CMOS chips; David Jäckel, Marta Lewandowska, Urs Frey, Sergey Sitnikov, Marie Engelene Obien and Riley Zeller-Townson for critical discussions; and the D-BSSE support staff for help with experiments.

## Additional information

### Funding

| Funder | Grant reference number | Author |
|---|---|---|
| H2020 European Research Council | Advanced Grant 267351'NeuroCMOS' | Andreas Hierlemann |
| H2020 European Research Council | Advanced Grant 694829 'neuroXscales' | Andreas Hierlemann |
| Schweizerischer Nationalfonds zur Förderung der Wissenschaftlichen Forschung | 205321_157092/1 | Andreas Hierlemann |
| Schweizerischer Nationalfonds zur Förderung der Wissenschaftlichen Forschung | Ambizione Grant PZ00P3_132245 | Douglas J Bakkum |
| Schweizerischer Nationalfonds zur Förderung der Wissenschaftlichen Forschung | Ambizione Grant PZ00P3_167989 | Felix Franke |

The funders had no role in study design, data collection and interpretation, or the decision to submit the work for publication.

### Author contributions

Milos Radivojevic, Conceptualization, Data curation, Software, Formal analysis, Validation, Investigation, Visualization, Methodology, Writing—original draft, Writing—review and editing; Felix Franke, Conceptualization, Software, Formal analysis, Methodology, Writing—review and editing; Michael Altermatt, Conceptualization, Software, Formal analysis, Validation, Investigation, Visualization; Jan Müller, Software, Methodology; Andreas Hierlemann, Conceptualization, Resources, Supervision, Funding acquisition, Project administration, Writing—review and editing; Douglas J Bakkum, Conceptualization, Resources, Software, Supervision, Funding acquisition, Methodology, Project administration, Writing—review and editing

### Author ORCIDs

Milos Radivojevic (iD) http://orcid.org/0000-0001-8828-9475

### Ethics

Animal experimentation: All experimental protocols were approved by the Basel Stadt veterinary office according to Swiss federal laws on animal welfare and were carried out in accordance with approved guidelines (Tierschutzgesetz TSchG, SR 455; Tierschutzverordnung TSchV, SR 455.1).

### Decision letter and Author response

Decision letter https://doi.org/10.7554/eLife.30198.010
Author response https://doi.org/10.7554/eLife.30198.011

## Additional files

### Supplementary files
• Transparent reporting form

DOI: https://doi.org/10.7554/eLife.30198.009

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
