## [Decision Letter]

Thank you for submitting your article "Tracking Individual Action-Potentials Throughout a Mammalian Axonal Arbor" for consideration by *eLife*. Your article has been reviewed by two peer reviewers, and the evaluation has been closely overseen by Reviewing Editor David Kleinfeld. Eve Marder serves as the Senior Editor. The reviewers have opted to remain anonymous.

The reviewers have discussed the reviews with one another and with the Reviewing Editor. We congratulate you on lovely experimental work that beautifully illustrates cell biophysics. The experimental tests of single-neuron computational ideas, such the notions of branch point failure (2 million records!) and jitter as a function of spike rate, are important. We request that you make changes to the manuscript that, with one exception, require only Discussion points. These will address the issues raised by the reviewers and the Reviewing Editor.

1) The statement in the third paragraph of the Introduction on the phototoxicity of voltage sensitive dyes is overstated. It has been extensively documented that action potentials can be monitored optically in the axonal arbor in brain slices without extensive averaging using voltage indicators. Please consider a revised statement.

2) It is probably incorrect to call cell culture of cortical neurons "cortical networks". The latter term implies a particular function in the brain which is certainly not maintained in culture.

3) The relationship between the amplitude of extracellularly recorded AP signals and the amplitude of the transmembrane voltage transients that represent AP should be included. Without clarification, the statement in the fifth paragraph of the Discussion that APs from the initial segments are 10-times larger in amplitude is misleading, as only the recordings of these signals are 10-times larger.

4) Please add images of the experimental setup, including neurons plated and grown on the MEA, as a supplementary material. Further, please clarify the extent of histological staining.

5) Much of the Discussion on the technique belongs in the Materials and methods section. This will streamline the manuscript.

6) There is prior art on optical measurements of signal propagation in processes of single cells from Fromherz (e.g., Fromherz and Muller, PNAS 1994), Kleinfeld (Stepnoski et al. PNAS 1991), and Zecević (Zecević, Nature 1996; Antić et al., J Physiol 2000; Djurisic et al., J Neurosci 2004; Foust et al., J Neurosci 2010) that bears on speed and pulse width, although none of the prior data sets are as strong as your current data. Some historical prospective would be useful.

7) Discussion of the computational aspects of your work could be strengthened, which would also ensure a larger audience. For example, the increased jitter with high spike rates is consistent with a growing level of inactivation of the Na-currents. You might also relate the changes in jitter of presynaptic cells to changes in temporal coincidence of PSPs in a common postsynaptic cell, and thus the reliability of readout. In this regard, we suspect that the effect of propagation delay is small, as the jitter is only ~0.1 ms, but the effect of jitter on synaptic release probability will still be significant through jitter in the height of the AP that effects the presynaptic voltage-dependent Ca-flux.

8) In support of a more thorough computational approach, a plot of the fractional jitter in peak extracellular voltage as a function of distance from the soma, much like the existing plots of temporal jitter as a function of distance from the soma (Figure 4, Figure 5), would be a welcome addition to the analysis in Results.

---

## [Author Response]

[…] 1) The statement in the third paragraph of the Introduction on the phototoxicity of voltage sensitive dyes is overstated. It has been extensively documented that action potentials can be monitored optically in the axonal arbor in brain slices without extensive averaging using voltage indicators. Please consider a revised statement.

We revised our statement in the third paragraph of the Introduction section and added references to prior art.

2) It is probably incorrect to call cell culture of cortical neurons "cortical networks". The latter term implies a particular function in the brain which is certainly not maintained in culture.

In response to this comment we edited the first paragraph of the Results section by replacing the term "cortical networks" by "cell cultures of cortical neurons".

3) The relationship between the amplitude of extracellularly recorded AP signals and the amplitude of the transmembrane voltage transients that represent AP should be included. Without clarification, the statement in the fifth paragraph of the Discussion that APs from the initial segments are 10-times larger in amplitude is misleading, as only the recordings of these signals are 10-times larger.

We edited the fourth paragraph of the Discussion section and added citation to Gold et al., 2007 to clarify this point. The amplitude of the signal at the AIS is now discussed in more detail in this paragraph and also in the fifth paragraph of the Discussion.

4) Please add images of the experimental setup, including neurons plated and grown on the MEA, as a supplementary material.

We complemented our supplementary material by adding Figure 1—figure supplement 1that shows the experimental setup and micrographs of neurons grown on the HD-MEA. The immunostaining procedure, which was used to visualize neurons presented in Figure 1—figure supplement 1 is described in the Materials and methods subsection “Immunocytochemistry”. Statements referring to Figure 1—figure supplement 1 have been introduced in the Results section, first paragraph and in the Materials and methods subsection “HD-MEAs”.

Further, please clarify the extent of histological staining.

In the present study we performed live-cell visualization in order to illustrate our experimental approach (Figure 3, Figure 4, Figure 5 and Figure 2—figure supplement 1) as well as to demonstrate that spatiotemporal distribution of recorded signals matches neuronal morphology (Figure 1, Figure 2, Figure 3, Figure 4, Figure 5 and Figure 2—figure supplement 1). The latter is also supported by our previous investigations, where we used live-cell visualization to verify that HD-MEAs can be used to track axonal AP propagation (Bakkum et al., 2013; Radivojevic et al., 2016). In response to this comment we edited the manuscript by adding additional information in the Materials and methods subsection “Live imaging and microscopy” and referenced our prior publications.

5) Much of the Discussion on the technique belongs in the Materials and methods section. This will streamline the manuscript.

We agree with the reviewers that certain parts of the Discussion section, in particular those referring to the template matching method, belong to the Materials and methods section. In response to this comment we edited the manuscript by moving parts describing the template matching method from the Discussion section to the Materials and methods subsection “Template-matching-based AP detection”.

6) There is prior art on optical measurements of signal propagation in processes of single cells from Fromherz (e.g., Fromherz and Muller, PNAS 1994), Kleinfeld (Stepnoski et al. PNAS 1991), and Zecević (Zecević, Nature 1996; Antić et al., J Physiol 2000; Djurisic et al., J Neurosci 2004; Foust et al., J Neurosci 2010) that bears on speed and pulse width, although none of the prior data sets are as strong as your current data. Some historical prospective would be useful.

We edited the third paragraph of the Introduction section and added references to prior publications. Please also see our reply to comment 1.

7) Discussion of the computational aspects of your work could be strengthened, which would also ensure a larger audience. For example, the increased jitter with high spike rates is consistent with a growing level of inactivation of the Na-currents. You might also relate the changes in jitter of presynaptic cells to changes in temporal coincidence of PSPs in a common postsynaptic cell, and thus the reliability of readout. In this regard, we suspect that the effect of propagation delay is small, as the jitter is only ~0.1 ms, but the effect of jitter on synaptic release probability will still be significant through jitter in the height of the AP that effects the presynaptic voltage-dependent Ca-flux.

We thank the reviewers for pointing this out and added additional text to the eighth paragraph of the Discussion section. In particular, we point to the impact of jitter on the postsynaptic computation and discuss possible other sources of variability.

The point the reviewers raise with respect to amplitude variability is a bit more intricate, and we refer to our response to the next reviewer comment.

8) In support of a more thorough computational approach, a plot of the fractional jitter in peak extracellular voltage as a function of distance from the soma, much like the existing plots of temporal jitter as a function of distance from the soma (Figure 4, Figure 5), would be a welcome addition to the analysis in Results.

The reviewers raise the point that the amplitude of the action potentials has an influence on the height of the PSP. In this study, we do not have access to the PSPs but we could measure the amplitudes of the spikes. In the manuscript, we show the change of the spike waveforms with stimulation number and the change of spike amplitudes as a function of distance from the soma. However, we did not analyze the variability of the spike amplitudes across individual spikes. The reviewers suggest including an analysis pertaining to the standard deviation of spike amplitude as a function of electrode and stimulation number. Author response image 1 and Author response image 2 show such an analysis for the neuron shown in Figure 5 of the main text. As can be seen, the standard deviation of the spike amplitude seems to be fairly stable for each electrode and over stimulation number. It does increase slightly, but only by 7% on average from the 1^st^ stimulation to 100^th^ stimulation. The standard deviation of the spike amplitude is a function of the variability of the spike waveform plus the variability due to noise. The larger the spike amplitude, the smaller is the relative influence of the noise. The reviewers asked for a plot showing the fractional amplitude jitter (standard deviation). This is shown in Author response image 2. In both figures we see only a very small effect of stimulation number on the standard deviation of the spike amplitude, independent of the electrode (and, therefore, the average amplitude of the spike).

**Author response image 1. respfig1:** Amplitude variability as a function of stimulation number. The plot shows the standard deviation of the spike amplitudes of the neuron shown in Figure 5 of the main text as a function of stimulation number (1^st^ to 100^th^). The standard deviation is computed over 39 repetitions for each individual electrode (grey lines). The black line shows the mean over all electrodes; the green line the linear regression to the mean. The average standard deviation of amplitudes for periods of noise is shown in red as a comparison. The average standard deviation of spike amplitudes increases with stimulation number (7% from 1^st^ stimulation to 100^th^ stimulation), but the increase is small.

**Author response image 2. respfig2:** Fractionalamplitude variability as a function of stimulation number. The plot shows the standard deviation of the spike amplitude, divided by the spike amplitude for each electrode, of the neuron shown in Figure 5 of the main text.

Considering the fact that the extracellular amplitude is only a proxy for the intracellular amplitude (see also the added text to the Discussion section), we find it hard to draw any strong conclusions from this analysis and do not think that such a plot would significantly add to the main message of our paper and would prefer to not include it in the main text.

While performing this analysis we noticed, however, that the way to compute the standard deviation of the arrival time jitter could be improved. Instead of a linear regression of the standard deviation of the arrival times, we compute now the linear regression of the variances of the arrival times against the axial distance. This had only a small impact on the reported values, but it significantly changed the line fits in Figure 4 and Figure 5, which are now square-root functions instead of linear functions. The reason to fit the variance against distance instead of the standard deviation is that, if the jitter introduced by each axonal segment is independent of that of the other segments, the jitter as a function of distance can be described by a white noise Gaussian process, so that the variance increases linearly with axial distance. The data seem to support this notion.